# Evaluation of strategies using simulation model to control a potential outbreak of highly pathogenic avian influenza among poultry farms in Central Luzon, Philippines

Roderick Salvador[1,2], Neil Tanquilut[3,4], Rosemarie Macmac[5], Kannika Na Lampang[6], Warangkhana Chaisowwong[6], Dirk Pfeiffer[7,8], Veerasak Punyapornwithaya[6]*

**1** College of Veterinary Science and Medicine, Central Luzon State University, Science City of Muñoz, Nueva Ecija, Philippines, **2** PhD Degrees Program, Veterinary Science, Faculty of Veterinary Medicine, Chiang Mai University, Chiang Mai, Thailand, **3** College of Veterinary Medicine, Pampanga State Agricultural University, Pampanga, Philippines, **4** School of Environmental Science and Management, University of the Philippines, Los Banos, Laguna, Philippines, **5** College of Resource Engineering Automation and Mechanization, Pampanga State Agricultural University, Pampanga, Philippines, **6** Veterinary Public Health and Food Safety Centre and Excellent Center of Veterinary Public Health, Faculty of Veterinary Medicine Chiang Mai University, Chiang Mai, Thailand, **7** Department of Pathobiology and Population Sciences, Royal Veterinary College, London, United Kingdom, **8** Centre for Applied One Health Research and Policy Advice, City University of Hong Kong, Hong Kong SAR, PR China

* veerasak.p@cmu.ac.th

## Abstract

The Philippines confirmed its first epidemic of Highly Pathogenic Avian Influenza (HPAI) on August 11, 2017. It ended in November of 2017. Despite the successful management of the epidemic, reemergence is a continuous threat. The aim of this study was to conduct a mathematical model to assess the spatial transmission of HPAI among poultry farms in Central Luzon. Different control strategies and the current government protocol of 1 km radius pre-emptive culling (PEC) from infected farms were evaluated. The alternative strategies include 0.5km PEC, 1.5km PEC, 2 km PEC, 2.5 km PEC, and 3 km PEC, no pre-emptive culling (NPEC). The NPEC scenario was further modeled with a time of government notification set at 24hours, 48 hours, and 72 hours after the detection. Disease spread scenarios under each strategy were generated using an SEIR (susceptible-exposed-infectious-removed) stochastic model. A spatial transmission kernel was calculated and used to represent all potential routes of infection between farms. We assumed that the latent period occurs between 1–2 days, disease detection at 5–7 days post-infection, notification of authorities at 5–7 days post-detection and start of culling at 1–3 days post notification. The epidemic scenarios were compared based on the number of infected farms, the total number of culled farms, and the duration of the epidemic. Our results revealed that the current protocol is the most appropriate option compared with the other alternative interventions considered among farms with reproductive ratio ($R_i$) > 1. Shortening the culling radius to 0.5 km increased the duration of the epidemic. Further increase in the PEC zone decreased the duration of the epidemic but may not justify the increased number of farms to be culled. Nonetheless, the no-pre-emptive culling (NPEC) strategy can be an effective alternative to

**Data Availability Statement:** All data files are available from the figshare database (accession number: DOI: 10.6084/m9.figshare.11987604).

**Funding:** The research fund was granted by Veterinary Public Health and Food Safety Centre and Excellent Center of Veterinary Public Health, Faculty of Veterinary Medicine Chiang Mai University, Chiang Mai, Thailand. The funders had no role in study design, data collection and analysis, decision to publish, or preparation of the manuscript.

**Competing interests:** The authors have declared that no competing interests exist.

the current protocol if farm managers inform the government immediately within 24 hours of observation of the presence of HPAI in their farms. Moreover, if notification is made on days 1–3 after the detection, the scale and length of the outbreak have been significantly reduced. In conclusion, this study provided a comparison of various control measures for confronting the spread of HPAI infection using the simulation model. Policy makers can use this information to enhance the effectiveness of the current control strategy.

## Introduction

Highly Pathogenic Avian Influenza (HPAI) is a major concern in the poultry industry. It is included amongst the list of notifiable terrestrial and aquatic animal diseases being monitored by the World Organization for Animal Health (OIE) [1]. Massive mortalities due to infection and eventual culling characterize HPAI epidemic. Globally, HPAI outbreaks caused the loss of approximately 120 million birds from January 2013 to June 2018 [2]. There were 16 million birds killed in the 1999–2000 Italy outbreaks. Around 30 and 17 million birds destroyed in the epidemics experienced in the Netherlands and Canada, respectively [3]. The H5N6 HPAI outbreak in the Philippines resulted in the culling of around 400,000 birds. The introduction of Avian Influenza in the country was confirmed by the Philippine government on August 11, 2017. Commercial poultry farms in barangays San Carlos and Santa Rita of San Luis, Pampanga were affected. After a week, cases were confirmed in Nueva Ecija, a province adjacent to Pampanga. A second wave of the outbreak was reported on November 12, 2017 in Cabiao, Nueva Ecija. A total of 23 farms were identified to have bird flu infection in the two provinces from July to November, 2017. Sixteen farms were in Pampanga and seven in Nueva Ecija. Eight towns in Pampanga were affected (Apalit, Bacolor, Candaba, Lubao, Mabalacat, San Luis, San Simon, Sto.Tomas) and four towns in Nueva Ecija were included (Cabiao, Jaen, San Isidro, Zaragosa). During the outbreak, fifteen layer chicken farms, 6 duck farms and 2 quail farms were infected. An outbreak investigation identified key factors affecting the spread of the disease among farms including low biosecurity, having multiple species reared on the farm, no outbreak monitoring system, inability to identify the disease and uncontrolled trade in poultry commodities.

The disease is zoonotic with a high case fatality rate (CFR). The high CFR of HPAI raised concerns about the potential emergence of an influenza pandemic [4]. Since the start of the HPAI H5N1 epidemic in late 2003, 861 human cases and 455 deaths have been reported globally [1, 5, 6]. With its severe economic consequences and the pandemic potential of HPAI, an effective control program needs to be implemented upon outbreak confirmation. Generally, disease spread can be prevented in 3 ways by decreasing (1) the contact rate between animals through isolation or separation, (2) infectiousness of infected animals by culling or treatment, and (3) susceptibility of animals via vaccination [7, 8].

The baseline HPAI control strategy in the Philippines consists of culling of birds in infected farms, movement restrictions and epidemiological tracing of potential transmission contacts. It also includes the culling of all domestic birds within a 1 km radius from the infected property. Pre-emptive culling (PEC) was implemented in the HPAI epidemics in the Netherlands [9], Canada [3], and Italy [10]. Epidemic simulation results supported the implementation of the PEC as an additional control measure to better manage the outbreak. Considering the variation of poultry density across territories, the strategy identified to be the best against HPAI epidemic in other countries may not be suitable under Philippines' condition. Hence, there is

a need to identify an epidemiologically optimal strategy that is adapted to the local poultry industry. It is challenging to identify which measure can provide the best impact in managing HPAI epidemics. Without a strong reference in selecting the "best" approach, resistance from the affected stakeholders will be encountered during its implementation. For instance, the affected industry stakeholders may ask for a justification of deciding for a 1 km culling radius rather than a half km. In the absence of scientific evidence in support of a control measure, public pressure may dictate how an epidemic should be managed. To determine the best strategy, effects from different scenarios should be compared.

Mathematical modeling allows the evaluation of scenarios where real field experiments cannot be performed such as infecting a farm with HPAI and observing its spread. This method is a useful tool in simulating outbreaks of infectious diseases [11]. Epidemic simulation models are used to describe the pattern of disease spread and how different control measures can change it. For instance, intervention strategies were simulated to strengthen the contingency plans and programs against foot and mouth disease in Spain [12], Texas, USA [13], and Korea [14]. For avian influenza, simulation models have been done in Great Britain [15], Netherlands [16], South Carolina, USA [17], and Italy [18]. In epidemiological studies of HPAI, mathematical modeling has been used to quantify transmission parameters [7, 19–21] and how the disease spread in populations [22–24].

In this study, the spread of HPAI in Central Luzon, Philippines was simulated to examine the effect of the various control strategies on the progression of the epidemic. We compared outcomes regarding the magnitude and duration of the epidemic among different control strategies. The findings from this study can be utilized to inform the policymaker for improving and enhancing the HAPI control programs.

## Materials and methods

The study was conducted in Central Luzon, Philippines. The Central Luzon region is composed of 7 provinces including Aurora, Bataan, Bulacan, Nueva Ecija, Pampanga, Tarlac, and Zambales (Fig 1). Based on data from the Philippine Statistics Authority, Central Luzon is the

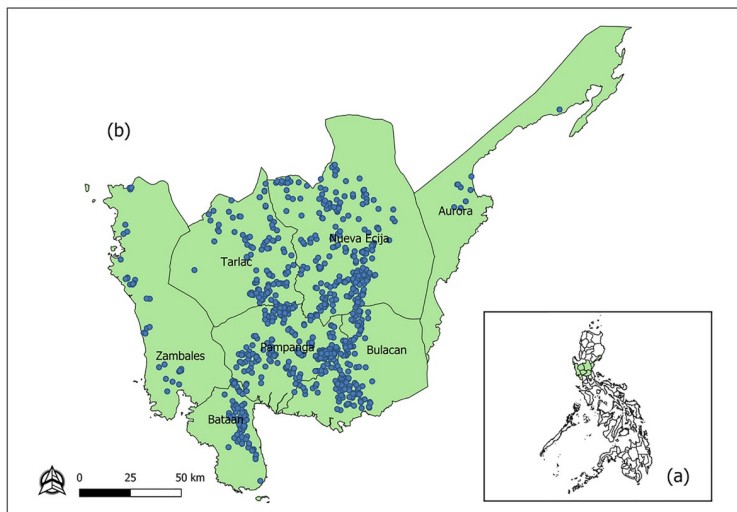

**Fig 1. Geographic distribution of poultry farms in central Luzon (CL) in 2018.** (a) map of Philippines showing the location of Central Luzon (CL) in green. (b) distribution of highly pathogenic avian influenza susceptible farms in the seven provinces of CL.

region with the highest number of ducks and duck farms in the country (32.02%) as of January 1, 2019. Moreover, the region shared 68.75% of the total stocks reared in commercial farms [25]. It also has the largest number of chickens with a 35.9% share of the country's total production over the same period [26].

The HPAI outbreak data was provided by the Bureau of Animal Industry (BAI), Philippines. The poultry farms included in this study were based on the registration records of the Provincial Veterinary Offices (PVO). Farm profiles (species-specific poultry population and farm coordinates) were compiled from farm visits conducted throughout the area. Each farm's maximum poultry capacity was used in the analysis.

Since registration is only required among commercial operations, therefore, small-scale poultry operations have not been included in the analysis. While 261 duck farms were registered, we believe that there were other commercial duck farms outside Pampanga. Although the number of unregistered quail and duck farms was unknown but this limitation may not be a serious issue. Compared to the operation of other species, a commercial quail farm does not require much space, and the transportation of quail eggs to the market can be done in concealed ways such as in vans or the boots of cars.

## Model formulation

The dynamics of disease spread in Central Luzon following the introduction of HPAI virus was estimated using a stochastic SEIR model. In this model, a farm's transition between each of the following states was modeled: S, susceptible; E, latently infected (infected but not yet infectious); I, infectious; and R, removed. Following the rationale of earlier studies [18, 27, 28], it was reasonable to define the farm as the epidemiological unit since the HPAI virus transmits rapidly between birds within an infected farm.

A transmission parameter called transmission kernel (*h*) was used to represent all forms of disease transmission between susceptible farms [29]. This approach has been used to HPAI [15, 18, 28, 30, 31] and other infectious animal disease models like FMD [29, 32, 33]. It refers to the infection hazard posed by an infectious animal from an infected farm to a susceptible animal of a susceptible farm. It is a function of the Euclidean distance (*r*) between farms. We adopted the kernel formulation (Eq 1) from the studies of Boender et al. [34, 35] and Dorigatti et al. [18]

$$h\left(r_{ij}\right) = \frac{h_0}{\left(1 + (r_{ij}/r_0)^{\alpha}\right)} \tag{1}$$

The $r_{ij}$ refers to the Euclidean distance between an infectious farm *j* and a susceptible farm *i*. The $h_o$ is the maximum hazard rate, occurring when $r_0 = 0$. The $r_0$ influences how far the hazard rate extends over distance. The $\alpha$ parameter influences the rate of decay in the hazard rate from the maximum. A dataset for the pairwise Euclidean distances (r) between farms was created. The distance between farms was calculated using the *distGeo* function of the geosphere package in R statistical software version 3.5.3 [36].

To calculate the likelihood function, the force of infection on a susceptible farm *i* at time *t* was given by Eq 2 as described in [34].

$$\lambda_i(t) = \sum_{j \neq i} h(r_{ij}) \, 1 \, [j \text{ is infectious}] \tag{2}$$

With $\lambda_i(t)$ as the force of infection or the cumulative hazard rate experienced by farm *i* on

day $t$, the probability that farm $i$ is infected on day $t$ is:

$$q_i(t) = 1 - e^{-\lambda i(t)} \tag{3}$$

The probability that farm $i$ remains uninfected up to day $t$ is:

$$r_i(t) = e^{-\sum_{s=1}^{t-1} \lambda_i(s)} \tag{4}$$

The force of infection ($\lambda$) towards susceptible farms $i$ on day $t$ depends on the cumulative number of infectious farms on day $t$, and the distances between the infectious farms and susceptible farms. With no infectious farm on day $t$, the force of infection ($\lambda$) is zero.

Given the risk of infection above, the log-likelihood function is given by:

$$\ell = -\sum_{k \in K} \sum_{t=1}^{t_{max}-1} \lambda_k(t) - \sum_{l \in \Lambda} \sum_{t=1}^{t_{cul,l}-1} \lambda_l(t) - \sum_{m \in M} \sum_{t=1}^{t_{inf,m}-1} \lambda_m(t)$$
$$+ \sum_{m \in M} log(1 - e^{-\lambda_m(t_{inf,m})}) \tag{5}$$

where the set $K$ contains all farms that remained uninfected and that were not culled; $\Lambda$ contains the farms that were not infected but that were culled (at times $t_{cul,l}$); $M$ contains the farms that were infected (at times $t_{inf,m}$). With Eq 5, the estimates for the transmission kernel parameter values ($h_o$, $r_0$ and $\alpha$) were calculated through maximum likelihood estimate (MLE) as described in [34] using the 2017 Philippines HPAI epidemic data. The *bbmle* package of the R program [36], was used for this procedure. Parameter values were presented in Table 1.

To construct the model, the factors used to account for the species-specific transmissibility were developed by Hayama (2015) [37], Keeling (2001) [38] and Dorigatti (2010) [18]. The number of animals in the farm is likewise added to the equation used by Keeling (2001) [38] and Hayama (2013) [37]. The inclusion of a species-specific transmission coefficient is considered important if there are variations in susceptibility and the occurrence of infections among different species, such as is the case with avian influenza. Unfortunately, no further parameters could be derived from this small dataset.

Following the procedure described by Boender et al. [34], the basic reproduction number was estimated for each poultry farm in Central Luzon using the spatial transmission kernel $h(r_{ij})$ and the stochastic infectious period (T) of farms $i$. The reproductive number of farms $i$, $R_i$, is given by

$$R_i = \sum_{j \neq i} (1 - \mathrm{E}\left[e^{-h(r_{ij})Ti}\right]) \tag{6}$$

**Table 1. Maximum likelihood estimation and 95% confidence intervals of the transmission parameters.**

|  | Estimate | 95% Confidence interval |
|---|---|---|
| $h_0$ (day$^{-1}$) | 0.0012 | 0.0001–0.1 |
| $r_0$ (km) | 3.4 | 1.001–10.0 |
| $\alpha$ | 1.4 | 1.001–5.0 |
| log-likelihood ($\ell$) | −206.99 | |
| AIC | 419.98 | |

The parameters $h_o$, $r_0$ and $\alpha$ determine the transmission kernel. The $h_o$ represents the maximum hazard rate. The $r_0$ influences how far the hazard rate extends over distance. The $\alpha$ parameter influences the rate of decay in the hazard rate from the maximum. The estimates for the transmission kernel parameter values and $\alpha$ were calculated through maximum likelihood estimate.

## Assumptions

Upon infection, a farm passes through a latency period of 1–2 days [18, 39–41]. In this stage, infected birds remain non–infectious. This is followed by the start of the infectious period in which birds become clinically ill. The detection of HPAI in the farm is assumed to be on days 5–7 based on mortality thresholds [42]. We used the mortality threshold because several studies indicated that the mortality threshold is the most reliable indicator of AI infection in poultry farms [43, 44].

Based on the response of farm managers to the survey, the authorities will be notified of the presence of the disease 5–7 days after detection. Culling of the infected farm is initiated 1–3 days after receiving the notification. According to the outbreak data, the BAI was notified about the occurrence of unusually high bird mortalities from the index farm in July, 2017. Culling was only initiated on August 2, 2017. The delay can be attributed to the required process of confirming for the presence of HPAI as this is the first time the country experienced HPAI outbreak. Beyond August 9, infected farms were culled 1–3 days post notification. The 2017 HPAI outbreak data is provided as supplementary. The days required for completion of the culling of infected farms after reporting were 1–3 days, depending on the size of the farm. Specifically, for farms with a population $\leq 50,000$, the culling needs 1 day; for farms with $50,000 <$ population $\leq 100,000$, the culling needs 2 days; for farms with population $> 100,000$, the culling needs 3 days. The infectious period lasts until the completion of culling. It is also assumed that birds die only from HPAI infection or culling. The bird population in each farm is assumed to be at its maximum capacity.

Poultry farms with reproductive ratio ($R_i$) $\geq 1$ were classified as high-risk farms. In contrast, farms with $R_i < 1$ have reduced the probability of transmitting HPAI to other farms. Thus, they can be managed without pre-emptive culling of neighboring farms [34]. The calculated $R_i$ for each farm included in this study is provided as supplementary material.

## Simulations

Simulation was done by randomly selecting a high-risk farm as the index. Using the *runif* command of the R software [36], 100 farms were selected to serve as index farms. An index farm was assumed to be at latent stage (E) at the beginning of the simulation. Each outbreak scenario was simulated using 1,000 iterations. Outbreak simulations were based on a discrete time step of 1 day. The simulation was performed using R Software with the *foreach* and *doParallel* packages [36].

The control strategies were listed in Table 2. The 1kmPEC was used as the reference strategy. An outbreak was seeded at each index farm in turn. Epidemic outcomes were summarized

**Table 2. Control strategies used when simulating HPAI spread in Central Luzon, Philippines.**

| Strategy | Description |
|---|---|
| 0.5kmPEC | Pre-emptive culling of farms within 0.5 km from an infected farm |
| 1kmPEC | Pre-emptive culling of farms within 1.0 km from an infected farm |
| 1.5kmPEC | Pre-emptive culling of farms within 1.5 km from an infected farm |
| 2kmPEC | Pre-emptive culling of farms within 2.0 km from an infected farm |
| 2.5kmPEC | Pre-emptive culling of farms within 2.5 km from an infected farm |
| 3kmPEC | Pre-emptive culling of farms within 3.0 km from an infected farm |
| NPEC | No pre-emptive culling with reporting between 5 to 7 days after detection |
| NPEC24 | No pre-emptive culling. Reporting to authorities is done within 24 hours after detection |
| NPEC48 | No pre-emptive culling. Reporting to authorities is done within 48 hours after detection |
| NPEC72 | No pre-emptive culling. Reporting to authorities is done within 72 hours after detection |

as the total number of infected farms, the total number of depopulated farms, and epidemic duration. The total number of depopulated farms refers to the summation of infected farms and the pre-emptively culled farms. It is assumed that an infected farm is being depopulated. Epidemic duration refers to the difference between the dates of the first infection and final culling. Spatial maps showing the progression of the epidemic for each strategy were generated using QGIS [43] to visually present their impact on HPAI outbreaks starting with a single infected farm.

Epidemic size and duration were evaluated for normality using the Shapiro-Wilk test. The Kruskal -Wallis test was used to compare the epidemic outcomes. The level of statistical significance was set at $P < 0.05$. Input values for the different parameters were changed in the sensitivity analysis to evaluate their influence on the modeling results.

## Results

The Shapiro-Wilk test showed that the simulation data was not normally distributed. Therefore, the Kruskal Wallis test was performed. Tables 3–5 showed the results of different control strategies. The spatio-temporal progression of the HPAI epidemic under different control strategies were presented in Figs 2 and 3.

**Table 3. Comparison of the number of infected poultry farms estimated for the different control strategies.**

| Strategies | Median | Mean | 95% Confidence Interval | |
|---|---|---|---|---|
| 1kmPEC | 8 | 11 [e] | 9 | 13 |
| 0.5kmPEC | 28 | 28 [f] | 23 | 32 |
| 1.5kmPEC | 5 | 6 [g] | 5 | 7 |
| 2kmPEC | 4 | 5 [ghij] | 4 | 6 |
| 2.5kmPEC | 4 | 5 [gij] | 4 | 5 |
| 3kmPEC | 4 | 4 [gj] | 4 | 5 |
| NPEC | 268 | 230 [bc] | 206 | 253 |
| NPEC24 | 5 | 72 [defgh] | 51 | 92 |
| NPEC48 | 328 | 252 [a] | 222 | 282 |
| NPEC72 | 275 | 242 [b] | 218 | 266 |

Means with different superscripts in each column were statistically different.

**Table 4. Comparison of the total number of poultry farms culled estimated for the different control strategies.**

| Strategies | Median | Mean | 95% Confidence Interval | |
|---|---|---|---|---|
| 1kmPEC | 118 | 134 [j] | 117 | 150 |
| 0.5kmPEC | 176 | 141 [hj] | 121 | 162 |
| 1.5kmPEC | 135 | 142 [gj] | 129 | 154 |
| 2kmPEC | 185 | 179 [f] | 168 | 190 |
| 2.5kmPEC | 199 | 201 [de] | 191 | 211 |
| 3kmPEC | 220 | 203 [d] | 191 | 216 |
| NPEC | 268 | 230 [bc] | 206 | 253 |
| NPEC24 | 5 | 72 [k] | 51 | 92 |
| NPEC48 | 328 | 252 [a] | 222 | 282 |
| NPEC72 | 275 | 242 [b] | 218 | 266 |

Means with different superscripts in each column were statistically different.

**Table 5. Comparison of the duration (days) of HPAI epidemic scenario estimated for the different control strategies.**

| Strategies | Median | Mean | 95% Confidence Interval | |
|---|---|---|---|---|
| 1kmPEC | 33 | 39 [e] | 35 | 44 |
| 0.5kmPEC | 64 | 56 [d] | 50 | 63 |
| 1.5kmPEC | 27 | 30 [fgh] | 27 | 33 |
| 2kmPEC | 25 | 27 [gh] | 24 | 29 |
| 2.5kmPEC | 25 | 26 [hi] | 23 | 28 |
| 3kmPEC | 24 | 24 [i] | 21 | 26 |
| NPEC | 125 | 117 [ab] | 106 | 128 |
| NPEC24 | 28 | 57 [cdefgh] | 46 | 68 |
| NPEC48 | 125 | 111 [b] | 100 | 123 |
| NPEC72 | 141 | 123 [a] | 112 | 133 |

Means with different superscripts in each column were statistically different.

The reference strategy (1kmPEC) generated an epidemic scenario with 134 (95% CI: 117–150) culled farms including 11 (95% CI: 9–13) infected poultry farms representing to 0.95% of the total number of commercial poultry farms in the region (n = 1,151). This can be interpreted that there are 584,595 infected birds. The epidemic was estimated to stop after 39 (35–44) days.

Lowering the pre-emptive culling radius by 0.5 km significantly increased the number of infected farms (n = 28; p = 0.00) and duration of the epidemic (n = 56 days; p = 0.00). Nonetheless, the total number of culled farms was comparable (p = 0.92). In contrast, increasing the radius to 1.5 km significantly reduced the number of infected farms (n = 6; p = 0.00) and the duration of the epidemic (n = 30; p = 0.00). The total number of culled farms is likewise comparable with the estimates in the reference strategy. The strategies with wider PEC radii

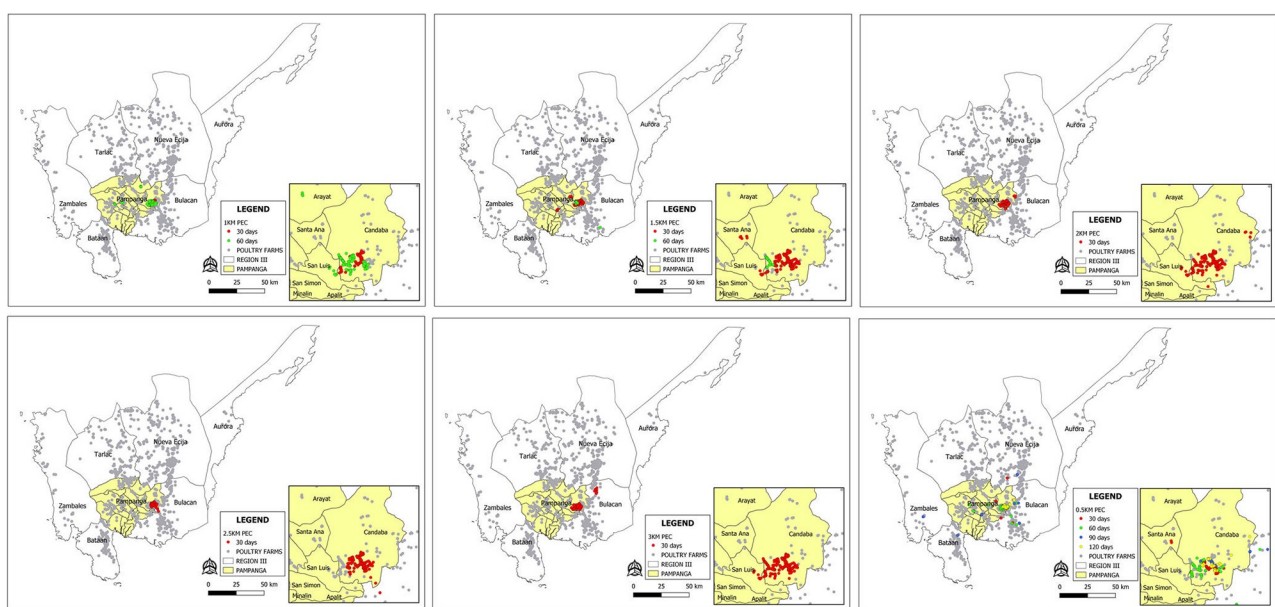

**Fig 2. Comparison of the spatio-temporal progression of HPAI epidemic managed with pre-emptive culling at various radius from infected farms.** Colored dots represent the totality of infected and pre-emptively culled farms according to the duration of the epidemic.

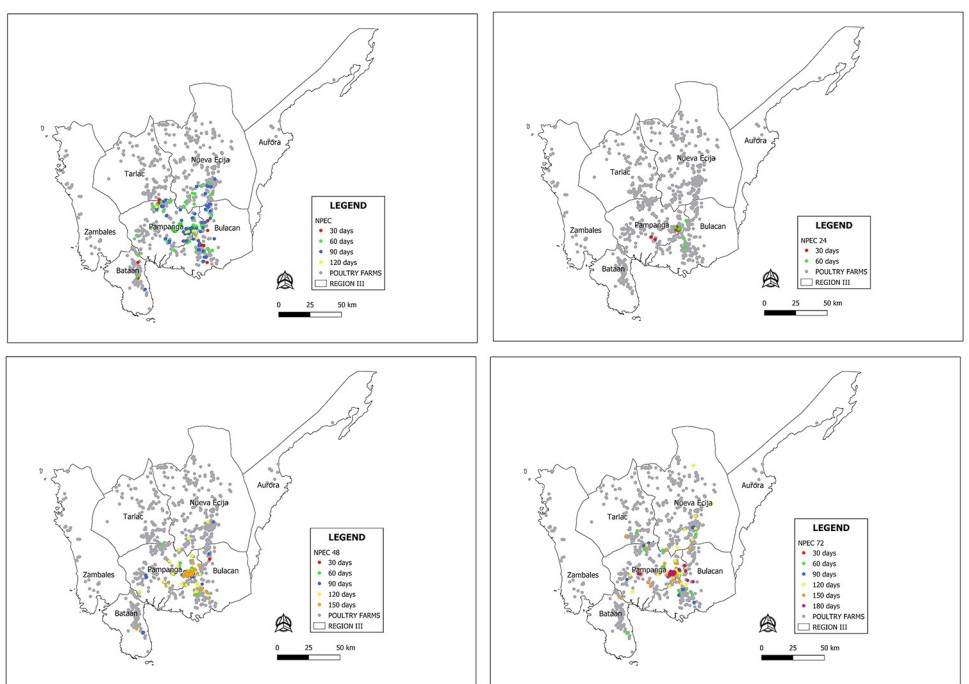

**Fig 3. Comparison of the spatio-temporal progression of HPAI epidemic managed with no pre-emptive culling at various reporting period.** Colored dots represent infected farms according to the duration of the epidemic.

(2kmPEC, 2.5kmPEC, 3kmPEC) significantly reduced the number of infected farms to around 5 farms and the duration of the outbreak to less than a month. The total number of culled farms increased to as much as 200 farms.

With no pre-emptive culling implemented (NPEC), the epidemic was estimated to infect 230 (95% CI: 206–253) farms in 117 (95% CI: 106–128) days. These estimates were significantly higher compared to the numbers of all strategies with pre-emptive culling. The number of infected farms under the NPEC24 scenario has estimates comparable to the pre-emptive culling from 0.5 to 2 km radii. The total number of culled farms (n = 72; 51–92) was significantly lower compared to the rest of the strategies evaluated. The duration of the epidemic was estimated at 57 (95% CI: 46–68) days. This result was comparable to the estimates for the strategies with pre-emptive culling implemented from 0.5 to 2.0 km radii.

## Spatio-temporal characteristics of HPAI epidemic

The spatial maps showing the progression of the HPAI epidemic from a single infectious farm for each strategy were presented in Figs 2 and 3. The generated risk map according to the farm level reproductive numbers is presented in Fig 4. The towns of Candaba and San Luis have at least 1.0 R*i* values. Thus, these towns are considered as high-risk zones (HRZ) (Fig 4). With 0.5kmPEC (Fig 2), the HPAI epidemic reached Nueva Ecija and Bulacan within 30 days. Within 120 days, the epidemic reached Zambales and Bataan provinces. With 1kmPEC (Fig 2), the epidemic affected only Pampanga. In 30 days, the epidemic spread to 1 municipality outside Candaba and San Luis. In 60 days, the epidemic spread to 3 municipalities outside the high-risk zone. With 1.5kmPEC (Fig 2), HPAI epidemic spread to 2 municipalities outside the HRZ in 60 days. The rest of the PEC strategies (Fig 2) limited the epidemic in the HRZ and stopped within 30 days.

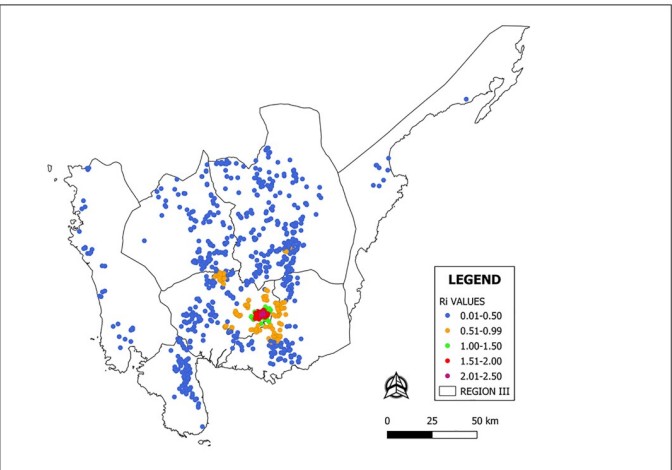

**Fig 4. Risk map of HPAI transmission in Central Luzon, Philippines; a) map showing the provinces in Central Luzon (b) map showing the towns within the high risk zone.**

For NPEC strategy (Fig 3), the epidemic spread in the provinces of Pampanga, Bataan, and Bulacan within 30 days. In 60 days, infection further spread to the Nueva Ecija province. The epidemic spread further to other municipalities in the 90 and 120- days scenario. The epidemic is limited in the municipalities of Pampanga for the NPEC24 (Fig 3) 30-days scenario. The epidemic reached Bulacan within 60 days. The NPEC48 (Fig 3) and NPEC72 (Fig 3) have similar spatio-temporal spread characteristics with NPEC affecting the provinces of Pampanga, Bataan, Bulacan, and Nueva Ecija.

## Sensitivity analyses

Sensitivity analyses were done for the latent period, detection time, notification time, and the parameter values for the transmission kernel. To minimize the number of comparisons, we compared the magnitude and duration of the epidemic with the reference scenario only.

Extending the latent period up to 5 days does not have any significant impact on the results we obtained from the 1–2 days latent period. A significant decrease in the mean duration ($p = 0.001$) and the number of infected farms ($p = 0.02$) were observed if HPAI infection is detected on the 3rd day post-infection. Detection on the 4rth day did not change the results of our assumptions. Though ideal, detection on the 3rd day is challenging. The clinical signs of avian influenza can also be observed among other endemic poultry diseases such as Newcastle disease and fowl cholera. The laboratory process to confirm HPAI infection likewise may surpass the 3rd day post-infection period. Earlier notification (1, 2, 3 days post-detection) significantly decreased the mean duration and the number of infected farms compared to our assumption (5–7 days). This is an important finding as the simple act of early reporting can limit the spread of the disease and protect the industry. We compared our derived transmission kernel estimates to the calculations of Boender et al. [34]. No significant difference was observed.

The scenario for the change in the time of reporting to authorities upon detection of HPAI presence in the farm was included in the variation of the NPEC strategy. Only the 24 hours reporting post-detection was found to change the estimates of the NPEC (Table 3).

## Discussion

The HPAI spread was simulated using a spatial transmission kernel. The transmission kernel was derived from the profiles of the 23 farms infected during the 2017 HPAI epidemic in the

Philippines. It is acknowledged that the results of this modeling should be used with caution, since only this single outbreak may be used to generate the transmission kernel, and farm locations and disease events could have been underreported.

In this study, the effectiveness of 10 different control strategies for managing HPAI epidemics were compared. Strategies include culling only the infected farm, and implementation of pre-emptive culling within various radial distances from the infected farm. The size (number of infected and pre-emptively culled farms) and duration of the epidemic were used for the comparisons. The estimates should not be interpreted as the exact number of farms that would be affected by a HPAI epidemic in Central Luzon. Simulation models can only provide a range of results as a reference for the likely size of the epidemic under each scenario.

The current government strategy (1km PEC) was used as the reference for comparison. Results showed that the current contingency program does not need modifications in high risk zones. The simulations indicate that there was no advantage in changing the PEC radiuses. Reducing the PEC radius to 0.5 km increased the number of infected farms, and the number of depopulated farms therefore increased. The same was observed for scenarios where the PEC radii were increased. The number of infected farms was reduced but these strategies required a wider area of pre-emptive culling. Eventually, the total number of farms culled increased. The duration of the epidemic estimated under the reference scenario can be further reduced to less than a month through the implementation of a wider PEC radius which will also mean that more farms will be depopulated. The implementation of pre-emptive culling from 2 to 3 km radius resulted to less than 30 days duration but increased the number of farms required to be culled. On the other hand, without pre-emptive culling (NPEC), the epidemic lasted longer (117 days) with significantly higher number of culled farms (n = 230; 206–253).

Livestock authorities face the decision-making dilemma for which outbreak response objective should be prioritized when selecting the optimal strategy. Ideally, the strategy should result in eradication of the disease as soon as possible with minimal numbers of animals culled. Nevertheless, these two objectives are competing. As shown by our results, the 3kmPEC was more effective in reducing the outbreak duration than the reference strategy. This would mean that trade restrictions could be lifted earlier benefitting exporters and local traders of poultry products. But the economic burden for farmers in the culling zone and the cost associated with the implementation of wide scale culling may be too high. The NPEC strategy prolongs the duration of the epidemic and resulted in a higher number of infected farms. The selection of the most appropriate strategy is therefore difficult and has to consider several variables and the weighting of each outbreak response objective varies between countries and within regions in countries.

During the HPAI epidemic, the BAI implemented the 1kmPEC strategy in Pampanga province. Our results showed that BAI's strategy was effective and is likely to have been economically acceptable, because changing the culling radius either way (0.5kmPEC and 1.5kmPEC) did not significantly reduce the size of the epidemic in terms of number of farms affected. The duration is reduced by 9 days if the 1.5kmPEC strategy is implemented. But this reduction is not enough to justify culling 8 additional farms. It is ideal to implement a strategy that can immediately stop disease spread. This objective is given much weight for public health diseases. Economics is another variable considered in selecting the best strategy against non-zoonotic diseases.

The outbreak observed in Nueva Ecija was managed using the NPEC strategy. The decision to shift to this approach may have been due to mounting pressure from the affected stakeholders against the implementation of PEC. Fortunately, our analysis on the farm level reproductive ratio ($R_i$) showed that farms outside the towns of San Luis and Candaba, Pampanga have $R_i < 1$. The $R_i$ refers to the average number of farms that an infected farm will infect

throughout its infectious period. If the $R_i$ is $\geq 1$, a self-sustaining epidemic is to be expected. Otherwise, the disease will not spread [35, 37, 44]. Thus, culling of the infected farm only is sufficient. There are no benefits of implementing pre-emptive culling if the $R_i$ is low. Thus, a key factor in identifying which control strategy is the most appropriate depends on where the outbreak occurs. The calculated farm level $R_i$ values presented in Fig 4 can be used in the future to decide if pre-emptive culling should be implemented in the event of HPAI outbreaks.

Technically, the use of the NPEC strategy outside the high-risk zone is generally favorable for the stakeholders. The high-risk areas identified in this study are concentrated only in Candaba and San Luis towns of Pampanga province. Hence, the NPEC strategy can be used in managing the HPAI outbreak outside these towns. However, using different strategies in different parts of a province can be perceived by poultry raisers whose premises were depopulated as an unfair treatment. To address this issue, the veterinary authority needs to develop a strong trust relationship with local farmers and veterinarians before HPAI outbreak events. Future modelling studies on this subject should also include a simulation scenario combining NPEC and pre-emptive culling with the calculated $R_i$ as a determinant. For instance, the control strategy can be initiated using pre-emptive culling and will automatically shift to the NPEC strategy for farms with $R_i \leq 1.0$.

The NPEC strategy can be implemented discretely. Thus, public panic can be avoided. During the early stage of the 2017 HPAI epidemic, pre-emptive culling strategy was used. The market price of poultry products declined drastically. This can be attributed to public concerns about consuming poultry because of the zoonotic potential of the disease [45–47]. Though the HPAI strain was later announced to be harmless, the stigma of the term" bird flu" heightened public fear to eat poultry products. Even poultry raisers outside the outbreak zone suffered economic losses because of the decreased demand. The market price of chicken started to normalize by October, 2017 [48]. During this period, the BAI kept on identifying other infected poultry farms outside the towns of San Luis and Candaba, Pampanga. At this stage of the outbreak, the NPEC strategy is being implemented. Despite the continuous implementation of the NPEC strategy, the public may not be able to monitor the progress of the outbreak. Eventually, the previous public fear stopped and start consuming poultry products again.

The NPEC24 resulted in the smallest outbreak in terms of the estimated number of affected farms (72; 51–92). The mean duration (57 days) was also comparable (p = 0.86) to the reference scenario (39 days). These numbers are likewise significantly reduced for the reference scenario (1 km PEC) if reporting to authorities is done within the first 3 days after the detection. These observations highlight the crucial role of livestock keepers in animal disease management. They are the vanguard of defense against disease outbreaks as they are in the position to lose the most. This means it should be a priority to incentivize farm managers to immediately notify the BAI upon detection of HPAI. Research such as conducted by Wright (2016) [49] is needed to identify locally relevant factors influencing farmers' reporting behaviors which will then allow developing strategies to increase engagement. Their results showed that perceptions of responsibility and trust are the main factors in determining if farmers will alert government to suspicious clinical signs of disease. Based on the results of the HPAI outbreak investigation, the BAI was only approached by the affected farm owners after poultry consultants had not been able to solve the observed unusual bird mortalities. Thus, the BAI should be recognized to have the expertise to diagnose and effectively manage poultry diseases. Of similar importance is enhancing the capability of farmers to detect the possible presence of avian influenza in their respective farms. The public awareness campaign used in the eradication efforts against Foot and Mouth Disease can be adopted. In the FMD program, a school on the air (SOA) radio training programs for pig producers was conducted. The training covered topics

on principles of disease recognition, reporting, diagnosis and control, plus improving hygiene and sanitation procedures [50].

A key part of effective outbreak management is lowering the possibility of disease transmission to susceptible farms. Neighbors of infected farms have a higher risk of becoming infected. Even with the implementation of movement restriction and increased biosecurity, between–farm transmission through untraced contacts ("neighborhood infections") still occurs, as has been shown during epidemics of avian influenza [51], foot-and-mouth disease (FMD) [38, 52], and classical swine fever [53–55]. To address this issue, farms around infected farms are usually either depopulated or vaccinated. The culling of farms located close to infected premises was implemented in the 2001 FMD epidemic in the United Kingdom. It was assumed that exposure to infection from neighboring infected premises occurred. Simulation analysis showed that this strategy resulted to fewer farms losing livestock [56]. Pre-emptive culling reduces local density of susceptible farms, thereby resulting in a decrease in the local reproductive ratio [56]. The 2016–2017 avian influenza epidemic in the Republic of Korea was managed through pre-emptive culling around infected premises with radius extending up to 3 km [57]. The management of the 2004 HPAI epidemic in Thailand likewise included pre-emptive culling. Initially, the radius from infected farms extend up to 5 km. It was later decreased to 1 km radius [58]. Likewise, pre-emptive culling involving high number of birds was performed during the HPAI epidemics in Vietnam [59].

Although the model supports the current HPAI contingency protocol for dealing with future outbreaks in high-risk areas, the implementation of pre-emptive culling is challenging. Stakeholders may not accept this approach due to its adverse economic effects. In fact, despite the epidemiological effectiveness of pre-emptive culling, large -scale depopulation of healthy animals has become increasingly unpopular in the public [60, 61]. Opposition from veterinarians, farm owners, and residents was reported by Bouma [44]. The evaluation of the policy for the 2001 FMD outbreak in the United Kingdom [62] indicated that "there was insufficient evidence to support the effectiveness of 3-km pre-emptive culling as a control procedure". The PEC strategy also requires enough manpower to meet the logistic demand of implementing the culling zone policy. Animals with high genetic values may be sacrificed. Biological pollution is also possible. The high number of carcass to be buried in an area can result to high volume of leachates. Burning as a method of carcass disposal can cause the release of dioxin in the atmosphere [28].

An alternative to the pre-emptive culling strategy is vaccination. It is more acceptable socio-politically [61, 63]. The outbreaks of FMD in Japan [64] and South Korea [65] were successfully managed using early vaccination. Vaccination against HPAI if combined with strict surveillance was shown to be effective in reducing the risk of further outbreaks [66, 67]. The susceptibility of vaccinated flocks is reduced because of the decreased transmission rate [1]. Vietnam [59], Hong Kong [68] and Italy [69] have included emergency vaccination in their contingency programs against HPAI.

The Philippines was successful in eradicating FMD without the implementation of pre-emptive culling. Control strategies implemented include quarantine and animal movement controls, strategic vaccination, surveillance and disease investigation, and enhanced public awareness with school on the air radio programs [50]. Poultry producers may likewise expect a more "acceptable" strategy for dealing with HPAI outbreaks. A mutually "acceptable" strategy can only be defined after the concerns of both the government and the poultry raisers are addressed. Factors to be considered in succeeding modelling research for HPAI in the Philippines may include vaccination, required resources such as manpower, and economic impact. A hybrid of PEC and NPEC strategies in which the management is dependent on the farm level reproductive number. For instance, the control strategy can be

initiated using pre-emptive culling and will automatically shift to the NPEC strategy for farms with $R_i \leq 1.0$.

## Conclusions

In this study, a spatial transmission kernel was used to explore the impacts of different HPAI outbreak control strategies. The results of these simulations provided estimates of epidemic size and duration for each of the control strategies considered. We also emphasized that the control strategy to be implemented is dependent on the calculated farm level reproductive number ($R_i$). With that information, it was possible to define high risk areas where the implementation of the reference strategy (1kmPEC) was optimal. In contrast, those occurring in low-risk areas could have been managed effectively using the NPEC strategy. Thus, the standard 1-km PEC protocol is not epidemiologically and socio-economically optimal for the whole region because of the variation in the reproductive number.

## Supporting information

**S1 Data. Dataset of farm location, animal population and animal species.**
(XLSX)

**S2 Data. Dataset of HAPI outbreaks in the Philippines in 2017.**
(XLSX)

**S3 Data. Data of reproduction number.**
(XLSX)

**S1 File. Results from sensitivity analysis.**
(DOCX)

## Acknowledgments

The authors would like to thank Dr. Ronnie Domingo, Dr Janice Sabagay-Garcia, Dr. Anthony Bucad of the BAI, the managers of the participating farms and everyone who extended their support in realizing this study.

## Author Contributions

**Conceptualization:** Roderick Salvador, Kannika Na Lampang, Warangkhana Chaisowwong, Veerasak Punyapornwithaya.

**Data curation:** Roderick Salvador, Neil Tanquilut, Rosemarie Macmac, Veerasak Punyapornwithaya.

**Formal analysis:** Roderick Salvador, Veerasak Punyapornwithaya.

**Funding acquisition:** Veerasak Punyapornwithaya.

**Methodology:** Roderick Salvador, Kannika Na Lampang, Warangkhana Chaisowwong, Dirk Pfeiffer, Veerasak Punyapornwithaya.

**Software:** Roderick Salvador, Veerasak Punyapornwithaya.

**Supervision:** Dirk Pfeiffer, Veerasak Punyapornwithaya.

**Validation:** Roderick Salvador, Dirk Pfeiffer, Veerasak Punyapornwithaya.

**Visualization:** Roderick Salvador, Veerasak Punyapornwithaya.

**Writing – original draft:** Roderick Salvador, Veerasak Punyapornwithaya.

**Writing – review & editing:** Neil Tanquilut, Rosemarie Macmac, Kannika Na Lampang, Warangkhana Chaisowwong, Dirk Pfeiffer, Veerasak Punyapornwithaya.

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
