## [Decision Letter · Decision Letter 0]

10 Jun 2020

PONE-D-20-13166

Evaluation of strategies using simulation model to control a potential outbreak of highly pathogenic avian Influenza among poultry farms in Central Luzon, Philippines

PLOS ONE

Dear Dr. Punyapornwithaya,

Thank you for submitting your manuscript to PLOS ONE. After careful consideration, we feel that it has merit but does not fully meet PLOS ONE’s publication criteria as it currently stands. Therefore, we invite you to submit a revised version of the manuscript that addresses the points raised during the review process.

The revised version should include sensitivity analyses of the most important parameters in the model. Moreover, assumptions need to be better justified, backed up with clear explanations, and supported by literature references. The text should be edited following the suggestions of the reviewers, and indeed, referring to the broader context can improve the paper and increase its impact. Please also consult the downloadable file with the comments of one of the reviewers.

We look forward to receiving your revised manuscript.

Kind regards,

Willem F. de Boer

Academic Editor

PLOS ONE

Journal Requirements:

2. Please upload a copy of Figure 14, to which you refer in your text on page 21. If the figure is no longer to be included as part of the submission please remove all reference to it within the text.

3. We note that Figures 1, 2, 7, 8, 9, 10, 11, 12, 13 in your submission contain [map/satellite] images which may be copyrighted. All PLOS content is published under the Creative Commons Attribution License (CC BY 4.0), which means that the manuscript, images, and Supporting Information files will be freely available online, and any third party is permitted to access, download, copy, distribute, and use these materials in any way, even commercially, with proper attribution. For these reasons, we cannot publish previously copyrighted maps or satellite images created using proprietary data, such as Google software (Google Maps, Street View, and Earth). For more information, see our copyright guidelines: http://journals.plos.org/plosone/s/licenses-and-copyright.

1.    You may seek permission from the original copyright holder of Figures 1, 2, 7, 8, 9, 10, 11, 12, 13 to publish the content specifically under the CC BY 4.0 license.

Additional Editor Comments (if provided):

Reviewers' comments:

Reviewer's Responses to Questions

**Comments to the Author**

1. Is the manuscript technically sound, and do the data support the conclusions?

Reviewer #1: Yes

Reviewer #2: Yes

2. Has the statistical analysis been performed appropriately and rigorously? 

Reviewer #1: Yes

Reviewer #2: Yes

3. Have the authors made all data underlying the findings in their manuscript fully available?

Reviewer #1: Yes

Reviewer #2: Yes

4. Is the manuscript presented in an intelligible fashion and written in standard English?

Reviewer #1: Yes

Reviewer #2: Yes

5. Review Comments to the Author

Reviewer #1: I think this is a very interesting work. The author provided a comparison of various control measures for confronting the spreading of HPAI infection. Finding an optimal control measure, which means to minimize the number of infected farms and minimize the number of culled birds as much as possible, is a very important topic. And policymakers need this information in making decisions and taking actions. The author used real HPAI outbreak information from Philippine’s farms, to test the effects of various control measures, which no doubt can provide some evidences in policymakers’ decision making.

But, I think this manuscript can be significantly improved after some modifications. For example, the authors used 8 figures to illustrate the spatial distributions of outbreaks under different control measures. These figures can be contained in one or two figures which use different color shades to illustrate the outbreak distribution. Moreover, figure 3-5 are repeated results from table 3-5. The authors should present these results in a more concise and compact way. Furthermore, the language in this manuscript can be, and should be improved for a smooth reading experience. Last but not least, the mathematical models should be explained with more details.

I am looking forward to read a revised version of this work.

Reviewer #2: Overall, the manuscripts is well written with a good discussion that describes the limitations of the study and the context within which this study can be used. However, the manuscript is very focused on the situation in the Philippines and could give additional information about control strategies in other countries that have dealt with HPAI outbreaks, to put the value of the study in a broader perspective. Additionally, it is not always clear how the authors have come to certain assumptions for the model. Some descriptions of the model and the assumptions are very short, which leaves rooms for interpretation and misconceptions. Please try to explain it more specifically and clearly, so the readers can easily understand the reasonings. Also add references to the papers that were used to substantiate the assumptions and were used as an example for the modeling approach.

Some parts of the manuscripts have a lot of detail, whereas others have little detail. I would recommend the authors to carefully read the manuscript again and check if everything has been explained clearly and remove parts that might not be relevant.

Abstract: a lot of words are dedicated to frame the study and only very few on the results, recommendations and conclusion. I would advise to shorten the first part and put more detail on model assumptions, results and recommendations as also described below.

Lines 43-44: I would recommend to add a summary of the different control strategies used in the model.

Lines 46-44: this is phrased strangely. I would suggest to rephrase this to something like: the model is based on the distance.

Lines 49: duration of what? Infection or outbreak? Be more specific in the descriptions and assumptions?

Lines 50-51: “Our results …. current protocol is enough”. Enough for what? Enough to control the disease within a certain time? More explanation is needed here.

Lines 63-65: The outbreak of HPAI in the Philippines in 2017 is mentioned here, however this is very short. It would be nice to have more information about the outbreak, i.e. how many farms, what type of farms were affected, were farms primarily infected through between farm transmission etc. Please provide a reference as well. This information is relevant as in the emphasis in this study is to model different strategies to reduce between farm transmission. This should be stated more clearly. Otherwise, one can get confused and wonder why no information is given on primary introductions caused by i.e. wild birds.

Lines 73-75: culling is not only a way to decrease the number of infected animals. Pre-emptive culling is a strategy to decrease the number of susceptible animals and should be mentioned, as this papers focuses on the pre-emptive culling strategy. Use a reference here.

Lines 87-93: a lot of emphasis is given here to the practical relevance of the study in order to convince stakeholders, however what is the scientific value? Please add some references as well.

Lines 94: “The only way to determine…” surely there are other ways as well, so please rephrase. Perhaps refer to other studies with a similar approach that have proven to be useful.

Lines 117-118: How accurate is the registration of the farms in the PVO? Are they all actual farms of do they also include backyard flocks? If so, is there any information on the role of the backyard flocks in the epidemic?

Lines 135-137: the abstract stresses that distance an important parameter: perhaps clarify how certain you are of the parameters in the transmission kernel: how accurate are these and did you also do sensitivity analyses on the assumptions for the transmission kernel. Please add references to papers that were used as an example for the transmission kernel (Boender et al 2007).

Lines 164-166: it would be useful here to have more information on the type of farms and when they were diagnosed (perhaps in supplementary). It would be interesting to have a comparison between the simulated data and the data of the actual outbreak (time of culling, time of notification, duration of outbreak etc).

Lines 168: table 1: please provide more detail about how the parameters were calculated in the superscript. It is mentioned in the manuscript and should also be mentioned here.

Lines 174 – 175: it is assumed that detection of HPAI infection is 5-7 days post infection and that notification of HPAI infection to authorities is 5-7 days post detection. In the sensitivity analysis it is described that in the modelling, the time to clinical onset is varied to 6-7 days post infection (lines 311-312) and that the latency period is prolonged to 3, 4 and 5 days, and that it is concluded that this resulted in less than 10% variation of outcome in the model. However, I wonder what would happen if these periods are shortened, especially the time to detection. In several countries detailed registration of mortality in poultry flocks has proven to be a sensitive tool for early detection of possible HPAI outbreaks, reducing the time to detection with several days, as well as time to notification. Improving detection methods would also make it more feasible for farmers to notify to the authorities earlier, which is shown by this study would also reduce the impact of the outbreak. This could be included in the discussion, at least as a recommendation as is has proven to be an important tool for early detection of HPAI infections. Also, it would be informative to have more information about how detection of infection occurs in the Philippines. Do farmers look for clinical symptoms of do they also take mortality into account?

Lines 176-177: how certain are you about these data? Is there a reference you can refer to?

Lines 185: please provide the data in supplementary information on the R0 of the farms.

Line 199: Only here the paper of Boender et al (2007) is first mentioned. This seems strange, as the approach and mathematical modelling clearly resemble the way it was proposed in the paper by Boender et al. (2007). Please refer to this paper, and other papers that were used as a reference or example for the modeling approach earlier in the manuscript.

Line 209: “The Spatial maps”, change to spatial maps.

Lines 204 + table 2: where are the control strategies based on? Why not use the default strategy (1kmPEC) and compare with earlier of later detection/culling or different values of the transmission kernel? It is mentioned in lines 307-316 that a sensitivity analysis has been performed, but the data has not been shown. Would it have added value to do a sensitivity analysis on the other assumptions as well (i.e. varying the infectious period) not only prolonging it, but reducing it?

Lines 259-264: How does this compare to the actual outbreak in 2017?

Lines 267: change “were” to “was”

Lines 357-360: it is mentioned here that the results from the study need to be used cautiously because only one outbreak provided information for the transmission kernel. Could you do a sensitivity analysis on the parameters of the transmission kernel too?

Lines 363-371: more information is given in this paragraph about the model and the assumptions that were made. Perhaps move this paragraph to the materials and methods section.

Lines 378: but how many farms were affected in the 2017 outbreak?

Lines 380-406: how to these findings relate to control of outbreaks in other countries, perhaps similar to the Philippines?

Lines 423: if the R0<1 the disease is less likely to spread, but you can still have some minor outbreaks due to stochasticity. Which assumptions were made to calculate the farm level R0 for the farms?

Lines 462-465: This seems an important finding, why not mention this more clearly in the abstract with recommendations to improve early detection and notification as well.

Lines 465-471: Isn’t it also important to provide recommendations on early detection and giving farmers tools to recognize the infection at an earlier stage?

In general: check the document for double spacing.

Figures 6-12: I was unable to read the text in figures because of poor quality. Please provide new figures with enough quality so the complete figure can be read and reviewed.

6. PLOS authors have the option to publish the peer review history of their article (what does this mean?). If published, this will include your full peer review and any attached files.

Reviewer #1: None

Reviewer #2: No

---

## [Author Response · Author response to Decision Letter 0]

13 Aug 2020

Response to editor

We created new figures (maps) based on suggestions from the reviewer. For all maps we created we used the resource from 

http://philgis.org/gis-data?fbclid=IwAR22DCFy2nAGi97CpuqliBzK6VgPeAdSPdIBrNE9Zgwyo0ONhHCU4up_dFo.

This site indicated that they support free geospatial data. This website also is public oriented GIS site.

Response to Comments

Reviewer #1

Reviewer #1: I think this is a very interesting work. The author provided a comparison of various control measures for confronting the spreading of HPAI infection. Finding an optimal control measure, which means to minimize the number of infected farms and minimize the number of culled birds as much as possible, is a very important topic. And policymakers need this information in making decisions and taking actions. The author used real HPAI outbreak information from Philippine’s farms, to test the effects of various control measures, which no doubt can provide some evidences in policymakers’ decision making.

But, I think this manuscript can be significantly improved after some modifications. For example, the authors used 8 figures to illustrate the spatial distributions of outbreaks under different control measures. These figures can be contained in one or two figures which use different color shades to illustrate the outbreak distribution. Moreover, figure 3-5 are repeated results from table 3-5. The authors should present these results in a more concise and compact way. Furthermore, the language in this manuscript can be, and should be improved for a smooth reading experience. Last but not least, the mathematical models should be explained with more details.

I am looking forward to read a revised version of this work.

Response: We have revised our manuscripts based on valuable comments and suggestion from reviewers. Thank you very much.

Based on your suggestion, Figures 3-5 were deleted. The 8 figures were reduced to just 2 figures using the duration of the epidemic as the reference in the variation of colors. Many points in this manuscript were revised.

Line Specific comments towards PONE-D-20-13166:

 2: “Influenza” —>”influenza”

Response: This was addressed in the title. Please see line 4. 

Line 44: “preemptive” is one single word, so please correct the “Pre-emptive” through the whole manuscript.

Response: Pre-emptive culling is now used throughout the manuscript. 

Line 49: “duration” should be more specific, as “pandemic duration”.

Response: epidemic duration or duration of the epidemic are now used in the manuscript. 

Line 52: remove “will”.

Response: this has been done. See line 68

Lie 53: “post detection”, I think you mean “after the detection“.

Response: post detection has been changed to after detection throughout the manuscript

Line 77: remove “all” and “present”.

Response: this has been addressed. See line 112

Line 78: I don’t think this is a grammatically correct sentence, please re-construct this sentence.

Response: The sentence has been modified. See lines 112-114. 

Line 84: Theoretically, culling should always be effective. So, it is “generalizable”. Please re-write this sentence.

Response: The sentence has been modified. See lines 118-120 

Line 87: “will”—>”can”.

Response: The sentence has been modified. See line 122

Line 92: “will be”—>”should be”.

Response: The sentence has been modified. See line 128

Line 99: “impact”—>”impacts”.

Response: The sentence has been modified. See lines 143-144

Line 119: “The farm profiles were compiled from farm visits”. You included more than 1,100 farms in this study, did you visited all of them? If you just visited part of these farms, then how did you compile the non-visited farms?

Response: All farms were visited. 

Line 137-138: These parameters were generated by MLE, but how and why? Please add more details and explanations.

Response: MLE was used as we adopted the modelling process described by Boender’s paper. More details were provided. Please see Lines 231-237

Line 142-166: These mathematical models should just be simply presented here. Please add more details and explanations.

Response: The mathematical models are now presented better. Please see lines 197-261

Line 193-197: If I understand the manuscript correctly, you randomly chose one farm as index farm, right? Then You totally chose 100 farms through your whole study. So under each of the scenario, you chose 1 farm as index, and run the simulation 1000 iterations, and then repeat this process for 100 times? Please make this part clearer

Response: That is what we did. 

Table 2: Please explain NPEC more clearly. What do you mean with “No culling with reporting within 24 hours…..”. It is not clear enough.

Response: The description has been modified for NPEC24, NPEC48, NPEC72. NPEC24 is now described as “No pre-emptive culling. Reporting to authorities is done within 24 hours after detection” 

Line 207: Please clearly define “epidemic duration”. Do you mean from the first infection occurring to last infected farm disappeared?

Response: Epidemic duration refers to the difference between the dates of the first infection and final culling. Please see lines 310-311. 

Line 262-263: you mentioned the percentage of duck farm, but is it important in your study? I didn’t mentioned farm type can influence anything before.

Response: The sentence “The majority of the infected were duck farms (78%)” was deleted

Line 268: “an additional 0.5 km”—>”1.5km”.

Response: The sentence has been modified. See line 349

Line 289-304: You mentioned lots of location names, but they are not necessary because (1) you didn’t label this location name in figure; (2) your simulation can not be interpreted as reality. I think you better to compare the distribution of outbreaks by overlapping them on the same figure. Remember, this is a qualitative not quantitative study. Moreover, using this names can only confuse your readers because they do not know where are the places.

Response: The maps were modified. Labels for the provinces were added. An insert map to show the location of the towns mentioned in the paper was included in Figure 4. The duration of the epidemic was overlapped in a single map for each intervention scenario. 

Line 372: “The results presented here…”—>”Our results…”

Response: The sentence was modified. See line 440-441

Line 381: how do you know that the current control measure is “adequent”? What’s the meaning of your “adequate”? I think this should be explained better, and more careful.

Response: The sentence was modified. We no longer used the term adequate. We instead rephrase the sentence into: “Results showed that the current contingency program does not need modifications in high risk zones.” See lines 450-451

Line 388-293: You compared the nr. of culled farms, pandemic duration and nr. of infected farms through the manuscript. But I think a more clearer and direct way to present (or compare) these three result is building a 3D plot. For example, z-axis is duration, x-axis is nr. of infected farms, y-axis is nr. of culled farms. I believe you can find a trade-off effect by making this plot.

The 3D plot was created. Please see the figure below;

We considered that this figure is not easy to be interpreted. We decided to use a simple table and the spatial maps to differentiate the impact of the different intervention strategies. 

Line 403-404: I don’t think you can select a “best” control strategy by doing this research, but more likely (or objectively) to show the trade-off effects among these control measure, by comparing their effects on duration, nr. of infected farms and nr. of culled farms. I think discussed about this trade-off is more objective and reasonable than “selecting the best strategy“, and concluding “current strategy is adequent”.

Your point is very interesting. For the main objective, we determine the best or the most appropriate strategy based on the total number of affected farms and the duration of the epidemic. The trade-off effects from different strategies were discussed in the study. The objective is mainly focus on determining of best strategy and the trade off effects obtained from several strategies were compared and discussed. Considering to your advice, we also use the word “the most appropriate” instead of “the best” when we refer to our model.

Line 408: Likewise, the location names doesn’t mean anything in current context.

Response: The sentence was modified. The location was still mentioned to emphasized the two approaches used by BAI in Pampanga and Nueva Ecija. A 1 kmPEC strategy was used in Pampanga. NPEC strategy was used in Nueva Ecija. Our results showed that BAI made the right move. Using the risk map as reference, the basic reproductive ration in Nueva Ecija is less than 1. Thus, NPEC is sufficient to manage the outbreak in this province. Pampanga on the other hand have farms with Ri ≥ 1. Thus, pre-emptive culling is recommended in this location. The location of the provinces is also included in the map for reference. See line 479. 

Line 409: Sometimes you use “BAI”, sometimes use “Bureau of Animal Industry”. Please be consistent.

Response: Bureau of Animal Industry was just used when BAI was introduced. The acronym BAI then was used throughout the manuscript. 

Line 412-416. Please re-organize this part, making it more clear.

Response: The statements were modified. See Lines 483-487 

Line 424. “.” is missing.

Response: a “.” has been added. See Line 491

Line 424-427. You mentioned that “R01 can still be a source of infection”, and then immediately, you also mentioned that culling in farms with such a small R0 brings no benefit. Please re-organize this sentence, making it more clear and concise.

Response: This section has been modified. See Line 493-498. 

Line 430. I think you have forgotten to upload you Fig. 14.

Response: Fig 14 is now Fig. 4. This figure shows the risk map based on the calculated farm level reproduction numbers. See line 498

Line 435. As Before, location names don’t mean anything here, unless you show them with labels in a figure.

Response: Location labels were added in the figures. 

Line 436: Again, this is a qualitative not quantitative study. You can compare the effects of various strategies, and discussed the trade-off, but it is far-fetched to make a conclusion about “safe” strategy. How do you define “safe”, and from which aspect you think this strategy is “safe”? Be careful with making a conclusion like this.

Response: The statement has been modified. The term “safe” has been changed. See lines 504-505

Line 448-449: Please re-organize this sentence, making it clearer and concise.

Response: The statement has been modified. See lines 515-517 

Line 457-458: I am confused here. I don’t think this can be assigned as “advantage” of NPEC. If the pandemic is not over, but the political decision makes people MISTAKENLY believe the pandemic is over, it may cause more problems. Please be careful with this opinion. I think you should clarify this part. 

Response: The statement has been modified. See lines 524-528 

Line 479-480. “Neighbors have a higher risk of becoming infected”. It seems to be a reasonable conclusion in your study, but you can expand the discussion a bit more. Actually, the transmission doesn’t only depend on distance, but also depends on connection. If two farms (one is susceptible, another one is infected) are far from each other, but very well connected by transportation, then the transmission may very much likely happen between them.

Response: Additional information about infection between neighboring farms was included. See lines 554-564. 

Disease transmission does not depend only on the distance between farms. Contact between farms is a major factor in disease transmission. Other modelling methods include other variables such as movement of products and personnel, frequency of deliveries, farm visits, level of biosecurity, probability of contamination, and other factors. The transmission kernel was used to represent these variables. The probability of transmission between farms decreases as distance increases. See lines 189-190. 

Line 494: “with”—>”in”.

Response: The statement has been modified. See line 578

Line497: remove “by Thrusfield (2005)”.

Response: This has been addressed. See line 581. 

Line 501: add example after “Biological pollution is also possible”.

Response: This has been addressed. Two examples of how mass culling can cause biological pollution were mentioned. See lines 584-587. 

Line 502-503: “…more acceptable sociopolitically…”

Response: This has been addressed. See line 590 

Line 504: You sometimes use “foot-and-mouth disease”, sometimes use “FMD”. Please be consistent.

Response: This has been addressed. Foot and Mouth Disease was used when FMD was introduced. FMD was then used throughout the manuscript. 

Line 515-516: please rephrase this sentence.

Response: The statement has been modified. See lines 603-609

Line 518: remove the “.”.

Response: This has been addressed. See line 608. 

Line 524-525: replace “to be… on” with “depends on”.

Response: This has been addressed. See lines 615-617. 

Line 529: What do you mean with the “region wide standard protocol”? Please make it clearer.

Response: Considering the variation in the farm level reproductive numbers, it is impractical to implement pre-emptive culling around a farm with Ri≤1. Thus, the standard 1 km PEC cannot be implemented throughout the region. See lines 620-622. 

Line 529-530: It seems there is a transition from your discussion to conclusion. You mentioned the current culling strategy is enough, but you concluded that culling is enough in high risk area, whereas culling is not optimal in low risk area in your conclusion part. The transition is not very well discussed. And you didn’t separate (or define) the high risk area and low risk area. I suggest to sort the ideas, and explain them step by step.

Response: Statements were rephrased to address this concern. It was now emphasized that the current protocol is for farms with Ri≥ 1. (See lines 62-67). The definition of high risk areas and low risk areas was provided. See lines 288-291.

Reviewer #2:

Reviewer #2: Overall, the manuscripts is well written with a good discussion that describes the limitations of the study and the context within which this study can be used. However, the manuscript is very focused on the situation in the Philippines and could give additional information about control strategies in other countries that have dealt with HPAI outbreaks, to put the value of the study in a broader perspective. 

Response: The experience of other countries that encountered HPAI epidemics are embedded in the discussion. For instance, on the discussion about the practice of pre-emptive culling, the management of Thailand, Vietnam, and Korea. In the discussion about vaccination against HPAI, the management of Vietnam, Hong Kong, and Italy was cited. 

For the revision, please see lines 565-571 and line 590-597.

Additionally, it is not always clear how the authors have come to certain assumptions for the model. Some descriptions of the model and the assumptions are very short, which leaves rooms for interpretation and misconceptions. Please try to explain it more specifically and clearly, so the readers can easily understand the reasonings. Also add references to the papers that were used to substantiate the assumptions and were used as an example for the modeling approach.

Response: This has been addressed. The modelling procedure is now better explained. The transmission kernel, force of infection, and the transmission kernel parameters have been defined (Please see line:192-202). Additional equations were added to better present the modelling process (line: 207-224). The basis for the assumptions are included (line 272-286). For instance, the basis for the 1-3 days period for the Bureau of Animal Industry to implement culling is from the outbreak data. Though, initially, culling implementation is at least 1 week post notification, the range of response duration on the later stage of the epidemic is 1-3 days. The assumption of 5-7 days for farm management to detect the presence of AI infection is based on the simulation made by Dorea. Suspicion of HPAI infection is related to mortality threshold. Likewise, references were cited to support these assumptions. 

Some parts of the manuscripts have a lot of detail, whereas others have little detail. I would recommend the authors to carefully read the manuscript again and check if everything has been explained clearly and remove parts that might not be relevant.

Abstract: a lot of words are dedicated to frame the study and only very few on the results, recommendations and conclusion. I would advise to shorten the first part and put more detail on model assumptions, results and recommendations as also described below.

Response: The abstract has been rewritten to accommodate this suggestion. 

Lines 43-44: I would recommend to add a summary of the different control strategies used in the model.

Response: The different control strategies are now included (line 52-55).

Lines 46-44: this is phrased strangely. I would suggest to rephrase this to something like: the model is based on the distance.

Response: The sentence was removed to give more emphasis on the results and recommendations

Lines 49: duration of what? Infection or outbreak? Be more specific in the descriptions and assumptions?

Response: We have revised. Please see line 62. The epidemic scenarios were compared based on the number of infected farms, total number of culled farms, and duration of the epidemic. 

Lines 50-51: “Our results …. current protocol is enough”. Enough for what? Enough to control the disease within a certain time? More explanation is needed here.

Response: The statement was rephrased. More explanation was made. Please see line 62-67.

Our results showed that the current protocol is the best option compared with the other alternative interventions considered among farms with Ri > 1. Shortening the culling radius to 0.5 km increased the duration of the epidemic. Further increase in the PEC zone decreased the duration of the epidemic but may not justify the increased number of farms to be culled. 

Lines 63-65: The outbreak of HPAI in the Philippines in 2017 is mentioned here, however this is very short. It would be nice to have more information about the outbreak, i.e. how many farms, what type of farms were affected, were farms primarily infected through between farm transmission etc. Please provide a reference as well. This information is relevant as in the emphasis in this study is to model different strategies to reduce between farm transmission. This should be stated more clearly. Otherwise, one can get confused and wonder why no information is given on primary introductions caused by i.e. wild birds.

Response: More details are now provided about the HPAI epidemic in the Philippines in 2017. The number and type of infected birds is provided. The details of the outbreak (date of notification, date of culling, coordinates, type of poultry) is provided as supplementary. The data came from the Bureau of Animal Industry (Please see: line 83-100). Other details of the outbreak came from the results of the outbreak investigation in which one of the authors (RTS) was a member of the investigating team. The report was submitted to the Food and Agriculture Organization (FAO) and was not published. 

How HPAI virus entered the Philippines was not established. Unusually high mortality were experienced by several farms as early as April, 2017. These cases were managed by company veterinarians as another wave of Newcastle disease (ND) outbreak. Central Luzon, Philippines was hit by ND epidemic in 2016. Being unable to address this concern, owners of affected farms finally notified the government by July, 2017. 

The trading of poultry between the provinces of Nueva Ecija and Pampanga contributed to the spread of the disease. Some of the affected farms are also owned by a single family. However, details like this are not included in the paper. 

The between farm transmission was set as a function of distance between an infectious and susceptible farm. The inclusion of the outbreak details may distract the reader on the concept of spatial transmission kernel. 

Lines 73-75: culling is not only a way to decrease the number of infected animals. Pre-emptive culling is a strategy to decrease the number of susceptible animals and should be mentioned, as this paper focuses on the pre-emptive culling strategy. Use a reference here.

Response: The statement was rephrased, and two references were cited (Line 107-110).

Lines 87-93: a lot of emphasis is given here to the practical relevance of the study in order to convince stakeholders, however what is the scientific value? Please add some references as well.

Response: The following paragraph was added. 

Mathematical modeling allows the evaluation of scenarios where real field experiments cannot be performed such as infecting a farm with HPAI and observing its spread. Hence, it is an effective tool in simulating outbreaks of infectious diseases[11]. Epidemic simulation models are used to describe the pattern of disease spread and how it can be changed by different control measures. Alternative intervention strategies were simulated to strengthen the contingency plans and programs against foot and mouth disease in Spain[12], Texas, USA[13], and Korea[14]. For avian influenza, simulation models have been done in Great Britain[15], Netherlands[16], South Carolina, USA[17], and Italy[18]. In epidemiological studies of HPAI, mathematical modelling has been used to quantify transmission parameters [7,19–21] and how disease spread in populations[22–24] . Please see Line 131-141.

Lines 94: “The only way to determine…” surely there are other ways as well, so please rephrase. Perhaps refer to other studies with a similar approach that have proven to be useful.

Response: The sentenced was rephrased to” To determine the best strategy, effects from different scenarios should be compared.”. line 128-129

Seven similar studies (comparison of alternative strategies against animal diseases) were cited. Please see line 134-141

For instance, intervention strategies were simulated to strengthen the contingency plans and programs against foot and mouth disease in Spain [12], Texas, USA [13], and Korea [14]. For avian influenza, simulation models have been done in Great Britain [15], Netherlands [16], South Carolina, USA [17], and Italy [18]. 

Lines 117-118: How accurate is the registration of the farms in the PVO? Are they all actual farms of do they also include backyard flocks? If so, is there any information on the role of the backyard flocks in the epidemic?

Response: The registration of farms in the Provincial Veterinary Offices was not so accurate. Only commercial farms are required to register. There is no registry for smaller poultry operations. This is one limitation of the study. There were no duck farms registered outside Pampanga province. Likewise, only two quail farms were registered. 

Please see line 170-177 for the revision.

The role of backyard flocks in the epidemic was not investigated. Backyard operations during the epidemic stopped either because of infection or pre-emptive culling. 

Lines 135-137: the abstract stresses that distance an important parameter: perhaps clarify how certain you are of the parameters in the transmission kernel: how accurate are these and did you also do sensitivity analyses on the assumptions for the transmission kernel. Please add references to papers that were used as an example for the transmission kernel (Boender et al 2007).

Response: We added the reference as suggested. Please see Lines 193-201. 

The parameters in the transmission kernel (h0, r0, and α ) were calculated through maximum likelihood estimate (MLE) following the procedure described by Boender et al. (2007). 

A sensitivity analyses on the assumptions for the transmission kernel was done and provided as supplementary. The parameters derived by Boender et al. (2007) was used. Results showed insignificant variation. Line: 409-411.

Lines 164-166: it would be useful here to have more information on the type of farms and when they were diagnosed (perhaps in supplementary). It would be interesting to have a comparison between the simulated data and the data of the actual outbreak (time of culling, time of notification, duration of outbreak etc).

Response: The actual outbreak data is provided as supplementary.

Lines 168: table 1: please provide more detail about how the parameters were calculated in the superscript. It is mentioned in the manuscript and should also be mentioned here.

Response: We add this paragraph to describe Table 1 as suggested. Please see line 241-245.

The parameters h_o, r_0, and α determine the transmission kernel. The h_o represents the maximum hazard rate. The r_0 influences how far the hazard rate extends over distance. The α parameter influences the rate of decay in the hazard rate from the maximum. The estimates for the transmission kernel parameter values and α were calculated through maximum likelihood estimate.

Lines 174 – 175: it is assumed that detection of HPAI infection is 5-7 days post infection and that notification of HPAI infection to authorities is 5-7 days post detection. In the sensitivity analysis it is described that in the modelling, the time to clinical onset is varied to 6-7 days post infection (lines 311-312) and that the latency period is prolonged to 3, 4 and 5 days, and that it is concluded that this resulted in less than 10% variation of outcome in the model. However, I wonder what would happen if these periods are shortened, especially the time to detection. In several countries detailed registration of mortality in poultry flocks has proven to be a sensitive tool for early detection of possible HPAI outbreaks, reducing the time to detection with several days, as well as time to notification. Improving detection methods would also make it more feasible for farmers to notify to the authorities earlier, which is shown by this study would also reduce the impact of the outbreak. This could be included in the discussion, at least as a recommendation as is has proven to be an important tool for early detection of HPAI infections. Also, it would be informative to have more information about how detection of infection occurs in the Philippines. Do farmers look for clinical symptoms of do they also take mortality into account?

Response: The result of the sensitivity analyses is included as supplementary. 

Extending the latent period up to 5 days does not have any significant impact on the results we obtained from the 1-2 days latent period. 

Please see lines 398-408. 

“A significant decrease on the mean duration (p=0.001) and number of infected farms (p=0.02) were observed if HPAI infection is detected on the 3rd day post infection. Detection on the 4rth day did not change the results of our assumptions. Though ideal, detection on the 3rd day is challenging. The clinical signs of avian influenza can also be observed among other endemic poultry diseases such as Newcastle disease and fowl cholera. The laboratory process to confirm HPAI infection likewise may surpass the 3rd day post infection period. HPAI infection among commercial poultry farms in the Philippines is suspected if a drastic increase in mortality is observed. Clinical signs associated to HPAI are likewise monitored. Earlier notification (1, 2, 3 days post detection) significantly decreased the mean duration and the number of infected farms compared to our assumption (5-7 days). This is an important finding as the simple act of early reporting can limit the spread of the disease and protect the industry. “

Lines 176-177: how certain are you about these data? Is there a reference you can refer to?

Response: The actual outbreak data is provided as supplementary for reference. 

Lines 185: please provide the data in supplementary information on the R0 of the farms.

Response: The calculated R0 for each farm is provided as supplementary.

Line 199: Only here the paper of Boender et al (2007) is first mentioned. This seems strange, as the approach and mathematical modelling clearly resemble the way it was proposed in the paper by Boender et al. (2007). Please refer to this paper, and other papers that were used as a reference or example for the modeling approach earlier in the manuscript.

Response: The paper of Boender et al. on modelling the FMD and HPAI epidemic in Netherlands were cited. The paper of Dorigatti was likewise mentioned in the model formulation section. Please see line 193-195.

Line 209: “The Spatial maps”, change to spatial maps.

Response: The recommendation has been addressed in the paper (Line 311).

Lines 204 + table 2: where are the control strategies based on? Why not use the default strategy (1kmPEC) and compare with earlier of later detection/culling or different values of the transmission kernel? 

Response: 

The control strategies are based on similar papers. The radius of the pre-emptive culling zone is varied to challenge the effect of the reference scenario. A sensitivity analysis including earlier detection is provided. 

The detection period was set at 3rd and 4rth day post infection since day 1 and day 2 are allotted for the latent period. Only the 3rd day detection has significantly decreased the epidemic size and duration. 

 Realizing detection as early as 3 days post infection is challenging because of three reasons: 1. Clinical signs of avian influenza can also be observed among other poultry disease including New Castle disease. 2. Mortality threshold that may trigger suspicion of HPAI infection may not be observed yet on the 3rd day post infection. 3. Laboratory processing to confirm diagnosis of avian influenza may take some time which eventually go beyond the 3rd day. 

The revision was presented in line 398-411.

It is mentioned in lines 307-316 that a sensitivity analysis has been performed, but the data has not been shown. Would it have added value to do a sensitivity analysis on the other assumptions as well (i.e. varying the infectious period) not only prolonging it, but reducing it?

Response: Results of the sensitivity analyses on the reduced period for detection, and notification is provided as supplementary. Both such variables affect the period of infectiousness. 

Lines 259-264: How does this compare to the actual outbreak in 2017?

Response: The simulated epidemic did not accurately match the actual outbreak. According to the reference strategy, there are 11 infected farms. The actual epidemic has 23. The difference can be attributed to several factors. The simulation used a single strategy (1 km PEC). Two strategy was used in the actual outbreak. Initially, the 1 km pre-emptive culling was used in Pampanga province. No pre-emptive culling (NPEC) strategy was used in Nueva Ecija. There was no record of farms pre-emptively culled. Hence, no comparison can be done for this variable. The duration of the simulated epidemic is shorter (39 days). The duration of the actual epidemic cannot be ascertained. The outbreak data only include the reporting date and the culling date. 

Despite this limitation, we believe that our model is sufficient to provide a good estimate on the epidemic size and duration for each strategy. 

Lines 267: change “were” to “was”

Response: We have changed it (line 348). 

Nonetheless, the total number of culled farms was comparable (p =0.92).

Lines 357-360: it is mentioned here that the results from the study need to be used cautiously because only one outbreak provided information for the transmission kernel. Could you do a sensitivity analysis on the parameters of the transmission kernel too?

Response: A comparison was done with the parameters of the transmission kernel derived by Boender for the Netherland’s HPAI epidemic. (Line 409-411)

Lines 363-371: more information is given in this paragraph about the model and the assumptions that were made. Perhaps move this paragraph to the materials and methods section.

Response: The paragraph you mentioned has been moved to the model formulation section. See Lines 247-254

Lines 378: but how many farms were affected in the 2017 outbreak?

Response: There were 23 farms identified to be infected with HPAI virus. The farms included in the pre-emptive culling strategy were not accounted. The data for the 2017 outbreak is included in the supplementary materials

Lines 380-406: how to these findings relate to control of outbreaks in other countries, perhaps similar to the Philippines?

Response: Researchers from other countries can follow same approach to simulate HPAI epidemic in their territory. 

Lines 423: if the R0<1 the disease is less likely to spread, but you can still have some minor outbreaks due to stochasticity. Which assumptions were made to calculate the farm level R0 for the farms?

Response: The reproductive number for each farm is based on the spatial kernel and the infectious period. We described the assumptions, please see line 256-261.

Lines 462-465: This seems an important finding, why not mention this more clearly in the abstract with recommendations to improve early detection and notification as well.

Response: We addressed this point in the abstract. Please see Lines 70-71.

Lines 465-471: Isn’t it also important to provide recommendations on early detection and giving farmers tools to recognize the infection at an earlier stage?

Response: The paragraph below was added in the discussion.

Of similar importance is enhancing the capability of farmers to detect the possible presence of avian influenza in their respective farms. The public awareness campaign used in the eradication efforts against Foot and Mouth Disease can be adopted. In the FMD program, a school on the air (SOA) radio training programs for pig producers was conducted. The training covered topics on principles of disease recognition, reporting, diagnosis and control, plus improving hygiene and sanitation procedures. Please see Lines 545-552.

In general: check the document for double spacing.

Response: We made a double spacing for the revision file.

Figures 6-12: I was unable to read the text in figures because of poor quality. Please provide new figures with enough quality so the complete figure can be read and reviewed.

 Response: We created new figures with a better quality. A set of large files with high resolution was published in figshare website. https://doi.org/10.6084/m9.figshare.12751445

---

## [Editor Report · Decision Letter 1]

25 Aug 2020

Evaluation of strategies using simulation model to control a potential outbreak of highly pathogenic avian influenza among poultry farms in Central Luzon, Philippines

PONE-D-20-13166R1

Dear Dr. Punyapornwithaya,

We’re pleased to inform you that your manuscript has been judged scientifically suitable for publication and will be formally accepted for publication once it meets all outstanding technical requirements.

Kind regards,

Willem F. de Boer

Academic Editor

PLOS ONE
---

## [Editor Report · Acceptance letter]

28 Aug 2020

PONE-D-20-13166R1 

Evaluation of strategies using simulation model to control a potential outbreak of highly pathogenic avian influenza among poultry farms in Central Luzon, Philippines 

Dear Dr. Punyapornwithaya:

I'm pleased to inform you that your manuscript has been deemed suitable for publication in PLOS ONE. Congratulations! Your manuscript is now with our production department. 

Kind regards, 

on behalf of

Dr. Willem F. de Boer 

Academic Editor

PLOS ONE